# Creating Data Standards to Support the Electronic Transmission of Compounded Nonsterile Preparations (CNSPs): Perspectives of a United States Pharmacopeia Expert Panel

**DOI:** 10.3390/children9101493

**Published:** 2022-09-29

**Authors:** Richard H. Parrish, Scott Ciarkowski, David Aguero, Sandra Benavides, Donna Z. Bohannon, Roy Guharoy

**Affiliations:** 1Department of Biomedical Sciences, School of Medicine, Mercer University, Columbus, GA 31207, USA; 2Pharmacy Quality & Safety, Michigan Medicine, Ann Arbor, MI 48109, USA; 3Medication Systems and Informatics, St. Jude Children’s Research Hospital, Memphis, TN 38105, USA; 4Handtevy Emergency Standards, Cumming, GA 30040, USA; 5Healthcare Quality and Safety, United States Pharmacopieal Convention, Rockville, MD 20852, USA; 6Division of Infectious Diseases, School of Medicine, University of Massachusetts, Amherst, MA 01655, USA

**Keywords:** compounding, drug, digital integration, electronic prescribing, health information interoperability, medication error, monograph, patient safety, standardization

## Abstract

The perspectives of the Compounded Drug Preparation Information Exchange Expert Panel of the United States Pharmacopeia (CDPIE-EP) on the urgent need to create and maintain data standards to support the electronic transmission of an interoperable dataset for compounded nonsterile preparations (CNSPs) for children and the elderly is presented. The CDPIE-EP encourages all stakeholders associated with the generation, transmission, and preparation of CNSPs, including standards-setting and informatics organizations, to discern the critical importance of accurate transmission of prescription to dispensing the final product and an urgent need to create and adopt a seamless, transparent, interoperable, digitally integrated prescribing and dispensing system benefiting of all patients that need CNSPs, especially for children with special healthcare needs and medical complexity (CSHCN-CMC) and for adults with swallowing difficulties. Lay summary: Current electronic prescription processing standards do not permit the complete transmission of compounded nonsterile preparations (CNSPs) from a prescriber to dispenser. This lack creates multiple opportunities for medication errors, especially at transitions of care for children with medical complexity and adults that cannot swallow tablets and capsules. The United States Pharmacopeia Expert Panel on Compounded Drug Preparation Information Exchange aims to reduce this source of error by creating ways and means for CNSPs to be transmitted within computer systems across the continuum of care. Twitter: Digitizing compounded preparation monographs and NDC-like formulation identifiers in computerized prescription systems will minimize error.

Some estimates place medical error as one of the leading causes of death in the United States [1], and one of the most concerning and significant errors occurs when a patient receives a discrepant prescription, that is, dispensed medication that differs from what was intended. According to the National Coordinating Council for Medication Error Reporting and Prevention, a medication error is defined as

“any preventable event that may cause or lead to inappropriate medication use or patient harm while the medication is in the control of the health care professional, patient, or consumer. Such events may be related to professional practice, health care products, procedures, and systems, including prescribing, order communication, product labeling, packaging, and nomenclature, compounding, dispensing, distribution, administration, education, monitoring, and use” [2].

Incorporating standards that foster e-prescribing (eRx) is a key tenet in the US federal government’s plan to reduce medication errors, increase implementation of electronic health records (EHR), and develop a nationwide EHR infrastructure for the United States. Many states and Medicare part D now require electronic transmission of controlled substances [3]. Soon, this mandate will be applied to all prescriptions [3]. While EHRs have minimized prescriber transcription errors, many pharmacy prescription processing systems do not interface with prescribing systems that would allow for seamless prescription transmission into pre-formatted data fields, both for finished dosage forms and compounded nonsterile preparations (CNSPs). Many children with special healthcare needs and medical complexity (CSHCN-CMC) utilize multiple pediatric CNSPs (pCNSPs) simultaneously on any given day. The likelihood for a medication error increases with the number of pCNSPs prescribed; the mis-selection of a solid oral dosage form with free-text instructions to prepare a pCNSP; the lack of adjudication and screening of pCNSPs through computerized decision support tools, including drug-drug interaction, allergy, and dose range checking; and the absence of complete pCNSPs on the patient’s medication list [4].

No data standards currently exist for the electronic transmission of pCNSP finished dose forms that include the liquid concentration or formulation ingredients. Manufactured products approved by the United States Food and Drug Administration (USFDA) as well as those found on the British National Formulary for Children [5] are listed and coded in computerized drug databases. Conversely, pCNSPs are not listed in any electronic repository that stores and allows their transmission into patient health record through eRx processing systems [6]. In addition, compounding monographs authorized by various professional bodies are not expressed in a digital format compatible with electronic transmission. This lack of digital integration, that is, the inability of electronic devices to exchange data or information through standardized algorithms, for both pCNSP identity and monograph recipes, can lead to significant errors in prescription generation and formulation [7,8]. Outpatient medication errors involving eRx for children have been estimated at between 6 and 13% of all transmissions [9]. In addition, liquid oral dosage forms (error rate = 21.6%) and younger patients (18.6%) are significant predictors of prescription errors in children [10]. Electronic clinical decision support (eCDS) for pCNSPs for medication reconciliation functions, drug–diagnosis mismatches, drug–drug interaction assessment, dose–range checking, and drug or constituent allergies, intolerances, and alerts do not exist for pCNSPs even though eCDS has been shown to reduce prescribing errors by 36% to 87% [11].

Errors related to a variety of medications for children have been reported in the literature. Studies conducted at the University of Michigan indicated that the number of pCNSP concentrations ranged from 1 to 9. The majority of pharmacies compounded more than three concentrations per pCNSP [12]. To mitigate errors, they developed one standard concentration for 100 commonly used pCNSPs and alternate concentrations for 4 of these pCNSPs [13]. Compounding errors are often attributable to the pCNSP concentration prepared for individual patients and disproportionately affect children [14]. One recent case report described an eight-fold clonidine overdose in a 12-year-old child who could not take tablets [15]. Two additional reports describe a one thousand-fold clonidine pCNSP overdose [16,17]. In fact, between 2003 and 2013, data from the American Association of Poison Control Centers identified that clonidine toxicity was most commonly associated with the need for intubation [18]. In a survey of 156 parents and caregivers of CSHCN-CMC, 10% of the patients were prescribed clonidine, topiramate, and zonisamide pCNSPs [19]. In another study of 232,240 pediatric patients with polypharmacy listed in an outpatient Medicaid database, 16.5% of patients in the high-risk category (i.e., ≥10 medications) were prescribed clonidine (38,920) [20]. There was no estimate given on the percentage associated with compounded clonidine liquid preparations. Other case reports detail significant patient morbidity involving flecainide [21] and atenolol [22] pCNSPs. These instances represent the “tip-of-the-iceberg” when it comes to identifying and reporting errors related to pCNSP eRx in the outpatient setting [1,23].

Indeed, variability in pCNSP orders presents an urgent need to codify uniform vocabulary standards at the systems level to mitigate dispensing errors. Even for existing digitized drug products, a cursory review of recent eRx evidence from across the world illustrates that (1) RxNorm contains variability in ingredient, strength, and dose form terminology [24], (2) dose strength (in mg/mL) might be the most common omission error [24,25], (3) manual product selection for eRx received in community pharmacies continues at a high rate [26], and (4) free-text note field entries are used frequently and inappropriately [27]. In the current process, prescribers cannot enter any USP-listed pCNSP from a drop-down menu as they can for USFDA-approved products assigned a National Drug Code (NDC) number [28]. The recently developed NDC directory pertains only to 503B outsourcing facilities, not traditional 503A compounding pharmacies, and non-pharmacy production entities are not required to assign NDCs to any pCNSP [29]. Most of these pCNSPs are entered in free text fashion or written on paper which generates even more potential errors [30]. While the most recent National Council on Prescription Drug Programs (NCPDP) SCRIPT standard enables the transmission of multiple ingredients formulated in a pCNSP from a pharmacy computer system to a third-party payor, it doesn’t configure the transmission of a standardized formulation and liquid concentration using any standardized and recognized compound formulation identifiers (CF-ID). Nor are prescribers informed about pCNSP ingredients contained within the preparation.

United States Pharmacopeia’s (USP) standards-setting role in the medication-use process is paramount to enhanced compatibility by creating digitized compendial monograph standards to aid in the management of medications of vulnerable populations, such as CSHCN-CMC and the elderly. USP’s Expert Panel (EP), Compounded Drug Preparation Information Exchange, is developing a set of encoding rules that would govern how pCNSPs are digitized and exchanged to improve medication management, especially at transitions of care. In this manner, transparency, interoperability, quality/safety, and reimbursement for pCNSPs would be optimized and systematic errors would be minimized. The primary outcome for encoding pCNSPs is to ensure CNSP formulation ingredients and compounding instructions are communicated to pharmacies, thereby decreasing the potential for medication errors caused by incomplete transmission of eRx information from authorized prescribers’ eRx platforms [31].

In summary, with the future mandate for electronic transmission of all prescriptions, the need for data standards to ensure safety and quality through semantic interoperability and transparency for pCNSPs is even more pressing. A significant unmet need exists to improve the standardization and transferability of pCNSP prescription data and information between healthcare settings, especially at transitions of care. Current EHR systems are highly variable in configuration and often not able to accommodate additional data elements that prescribers and pharmacists need to transmit. Interoperability between care settings is not optimal. Many pCNSP claims are rejected for payment. Significant challenges must be overcome to achieve standardization and transferability of CNSP data, including the wide variety of roles played by potential users, and varying perceived needs regarding data elements to be transmitted. Beginning the digitization process with pCNSPs, we believe that the creation of CF-IDs in an NDC-like format for all scientifically-validated sterile and nonsterile compounded preparations that both codifies a compounding monograph and allows bi-directional standardized concentration nomenclature transmission seamlessly across computer platforms, will address four salient objectives: (1) minimize prescription and formulation errors; (2) optimize patient safety; (3) expand pragmatic pediatric pharmacotherapy research opportunities related to pCNSP use; and (4) improve reimbursement for services rendered.

## Data Availability

Not applicable.

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
