# Peer review of "Creating Data Standards to Support the Electronic Transmission of Compounded Nonsterile Preparations (CNSPs): Perspectives of a United States Pharmacopeia Expert Panel"

_children, 2022, doi:10.3390/children9101493_

Round 1
Reviewer 1 Report
The concept electronic transmission of all prescriptions and data standards to ensure safety and quality is very good concept. What data standards are currently available and what methods they are used may be describe elaborately the authors.
Author Response
Reviewer 1:
The concept electronic transmission of all prescriptions and data standards to ensure safety and quality is very good concept. What data standards are currently available and what methods they are used may be describe elaborately the authors.
We thank the reviewer for the kind comments and insights about the current data standards for electronic transmission of CNSPs. At several points in the manuscript, we mention the lack of data standards for pCNSPs (lines 69-89 and 116-123). We will add the following sentence at line 69 to make it explicitly clear: “No data standards currently exist for the electronic transmission of pCNSP finished dose forms that include the liquid concentration or formulation ingredients.”
Reviewer 2 Report
The perspectives of the Compounded Drug Preparation Information presented by the Exchange Expert Panel of the United States Pharmacopeia (CDPIE-EP) resound well with their primary goal which is to attend to the urgent need to create and maintain data standards to support the electronic transmission of an interoperable dataset for compounded non-sterile preparations (CNSPs) for children and the elderly. Nowadays, clear electronic prescription processing standards are really of need, in order to avoid encountering patients with potential double/contraindications prescriptions.
The authors did well present the clinical significance of the topic. The paper is well written and sustained by the references. However, i do think that more content could have been added to stress out the scientific soundness of this paper. Nevertheless, based on the nature of this paper and of its structure, authors have done an important work.
Author Response
The perspectives of the Compounded Drug Preparation Information presented by the Exchange Expert Panel of the United States Pharmacopeia (CDPIE-EP) resound well with their primary goal which is to attend to the urgent need to create and maintain data standards to support the electronic transmission of an interoperable dataset for compounded non-sterile preparations (CNSPs) for children and the elderly. Nowadays, clear electronic prescription processing standards are really of need, in order to avoid encountering patients with potential double/contraindications prescriptions.
We appreciate the reviewer’s understanding of the existing need for data standards to electronically transmit complete pCNSPs.
The authors did well present the clinical significance of the topic. The paper is well written and sustained by the references. However, i do think that more content could have been added to stress out the scientific soundness of this paper. Nevertheless, based on the nature of this paper and of its structure, authors have done an important work.
We thank the reviewer for this insightful comment. The science of electronic prescription transmission continues to be an important area of research. We will add a cursory review of recent literature at line 111 with the following sentence: Even for existing digitized drug products, a cursory review of recent eRx evidence from across the world illustrates that (1) RxNorm contains variability in ingredient, strength, and dose form terminology [24], (2) dose strength (in mg/mL) might be the most common omission error [24,25], (3) manual product selection for eRx received in community pharmacies continues at a high rate [26], and (4) free-text note field entries are used frequently and inappropriately [27].
- Lester, C.A.; Flynn, A.J.; Marshall, V.D.; Rochowiak, S.; Rowell, B.; Bagian, J.P. Comparing the variability of ingredient, strength, and dose form information from electronic prescriptions with RxNorm drug product descriptions. J. Am. Med. Inform. Assoc. 2022, 29, 1471-1479. doi: 10.1093/jamia/ocac096.
- Murphy, A.P.; Bentur, H.; Dolan, C.; Bugembe, T.; Gill, A, Appleton R. Outpatient anti-epileptic drug prescribing errors in a Children's Hospital: an audit and literature review. Seizure. 2014, 23, 786-791. doi: 10.1016/j.seizure.2014.06.010.
- Panich, J.; Larson, N.; Sojka, L.; Wallace, Z.; Lokken, J. Assessing automated product selection success rates in trans-missions between electronic prescribing and community pharmacy platforms. J. Am. Med. Inform. Assoc. 2021, 28, 113-118. doi: 10.1093/jamia/ocaa259.
- Dhavle, A.A.; Yang, Y.; Rupp, M.T.; Singh, H.; Ward-Charlerie, S.; Ruiz, J. Analysis of Prescribers' Notes in Electronic Prescriptions in Ambulatory Practice. JAMA Intern. Med. 2016, 176, 463-470. doi: 10.1001/jamainternmed.2015.7786.